# Lack of evidence for associative learning in pea plants

Kasey Markel*

Plant Biology, University of California, Davis, Davis, United States

**Abstract** Gagliano et al. (Learning by association in plants, 2016) reported associative learning in pea plants. Associative learning has long been considered a behavior performed only by animals, making this claim particularly newsworthy and interesting. In the experiment, plants were trained in Y-shaped mazes for 3 days with fans and lights attached at the top of the maze. Training consisted of wind consistently preceding light from either the same or the opposite arm of the maze. When plant growth forced a decision between the two arms of the maze, fans alone were able to influence growth direction, whereas the growth direction of untrained plants was not affected by fans. However, a replication of their protocol failed to demonstrate the same result, calling for further verification and study before mainstream acceptance of this paradigm-shifting phenomenon. This replication attempt used a larger sample size and fully blinded analysis.

## Introduction

The temporal and spatial heterogeneity of any given environment poses a major challenge to all life on Earth. Accordingly, a common characteristic of living organisms is the capacity to respond to environmental cues. Many of these responses involve nuanced decisions integrating multiple cues or evaluating cue intensity, such as chemotaxis in bacteria and other microbes (*Wadhams and Armitage, 2004*). Even 'simple' organisms make decisions based on the integration of multiple cues or the intensity of particular cues. For example, unicellular *Dictyostelium discoideum* amoebae aggregate into multicellular slugs in the absence of food, and travel at precise, genetically determined angles in the direction of a light source (*Annesley and Fisher, 2009*).

Many complex environmental responses have recently been discovered in plants, including luring in animal predators to attack herbivores in response to herbivory (*De Moraes et al., 1998*), production of toxins in response to distress signals from kin (*Karban et al., 2013*), and dose-dependent responses to insufficient water (*Pandey et al., 2016*) or excess salt (*Julkowska and Testerink, 2015*) in the soil. These responses demonstrate an impressive capacity to detect a variety of environmental signals and respond accordingly. These instinctive behaviors have been sculpted over evolutionary time, and can be contrasted with learned behaviors that are developed within the lifetime of an individual as a result of their particular interactions with the environment. Individuals and groups of organisms leverage learning to exploit patterns in their environment over short timescales, and the capacity to do so frequently confers a selective advantage (*Morand-Ferron, 2017*). Learning is useful in any environment that contains exploitable regularities and patterns and for which the instincts of the organism have not evolved to perfectly exploit those patterns. Learning is ubiquitous among animals, from zebrafish (*Sison and Gerlai, 2010*) and insects (*Prokopy et al., 1982*) to nematodes (*Wen et al., 1997*), and recent work in *Caenorhabditis elegans* is beginning to unravel its molecular and genetic mechanisms (*Gyurkó et al., 2015*).

Over the last decade, there has been substantial debate in neurobiology and philosophy of mind regarding what kinds of organisms possess the capacity to learn (*Gagliano et al., 2016*; *Gagliano et al., 2018*; *Thellier, 2017*; *Gagliano, 2017*), along with thornier issues like cognition (*Garzón, 2007*; *Gross, 2016*; *Adams, 2018*; *Segundo-Ortin and Calvo, 2019*) and intelligence

*For correspondence:
Kaseymarkel@gmail.com

Competing interests: The author declares that no competing interests exist.

**eLife digest** Associative learning is a simple learning ability found in most animals, which involves linking together two different cues. For example, the dogs in Pavlov's famous experiment were trained to associate sound with the arrival of food, and eventually started salivating upon hearing the sound alone.

Plants, like animals, are capable of complex behaviors. The snapping leaves of a Venus fly trap or the sun-tracking abilities of sunflowers are examples of instinctive responses to environmental cues that have evolved over many generations. Whether or not plants can learn during their lifetimes has remained unknown.

A handful of studies have tested for associative learning in plants, the most convincing of which was published in 2016. In this study, pea plants were exposed to two signals: light, the plant version of dog food, and wind, equivalent to the sound in Pavlov's experiment. Just as dogs salivate in response to food, plants instinctively grow towards light, whereas air flow does not affect the direction of growth.

The plants were grown inside Y-shaped mazes and their 'selection' of one particular arm was used as a 'read-out' of learned behavior. The experiments trained growing plants by exposing them to wind and light from either the same direction or opposite directions. Once the plants were at the point of 'choosing' between the two arms, they were exposed to wind in the absence of light. Wind by itself appeared to influence the direction the trained plants took, with wind attracting plants trained with wind and light together and repelling plants trained with wind and light apart. Untrained plants remained unaffected, making random selections.

These observations were interpreted as the strongest evidence of associative learning in plants and if true would have great scientific and philosophical significance. Kasey Markel therefore set out to confirm and expand on these findings by replicating the 2016 study.

As many conditions as possible were kept identical, such as the training regime. The new experiments also used more plants and, most importantly, were done 'blind' meaning the people recording the data did not know how the plants had been trained. This ensured the expectations of the researcher would not influence the final results. The new study found no evidence for associative learning, but did not rule it out altogether. This is because some experimental details in the first study remained unknown, such as the exact model of lights and fans originally used.

This work demonstrates the importance of replicating scientific experiments. In the future, Markel hopes their results will pave the way for further, rigorous testing of the hypothesis that plants can learn.

---

(*Marder, 2013*; *Trewavas, 2017*). Some of these disputes center around particular terminology and language, but some basic empirical questions remain contentious, including the simple question: can plants learn? A series of experiments on associative conditioning have been carried out in plants since the 1960s, notably the efforts of *Holmes and Gruenberg, 1965*, *Haney, 1969*, *Levy et al., 1970*, and *Armus, 1970*. These studies generally failed to observe conditioning in plants, and those that concluded in favor of conditioning lacked sufficient controls to rule out other explanations, as reviewed by *Adelman, 2018*. *Gagliano et al., 2016* provide the most convincing report of conditioning in plants to date, but there are no published reports of replication, from the original lab or others. Unlike many previous experiments designed to test for learning in plants, Gagliano et al. lengthened the timelines of traditional animal learning experiments to be more appropriate for plant physiology. Training takes place over 3 days with stimuli lasting 60–90 min, with intertrial intervals of 60 min.

Associative learning is the phenomenon whereby an individual organism associates two stimuli, and thereafter uses one as an indicator for the other. In Ivan Pavlov's classic work, dogs were trained to associate the ringing of a metronome with the imminent arrival of food, and demonstrated the appropriate physiological response of salivation (*Pavlov, 1927*). Associative learning experiments attempt to pair an unconditioned stimulus for which there is an observable response to a conditioned stimulus which is originally neutral. Learning is demonstrated when presentation of the conditioned stimulus can elicit responses normally associated with the unconditioned stimulus. In

December 2016, Gagliano et al. published a report claiming plants have the capacity to learn adaptive behaviors during individual lifetimes using an experimental setup analogous to Pavlov's famous experiment (*Gagliano et al., 2016*).

## Results

### Experimental design

Gagliano et al. adapted Pavlov's paradigm for pea plants (*Pisum* sativum) using light as the unconditioned stimulus and wind as the neutral stimulus, which through training became the conditioned stimulus (*Gagliano et al., 2016*). Plants were grown inside a dark controlled growth chamber with individual PVC Y-mazes with blue LEDs and fans attached on each arm, and plant learning was inferred through the maze arm selection of the plants. This experimental setup is particularly suited to pea plants, which are topped during their early stages by a single tendril, which makes maze arm selection quite clear. See Materials and methods for growth conditions, germination protocol, and maze dimensions, which are unchanged in this study. *Figure 1—figure supplement 1* details the construction of the Y maze and the associated lights and fans required for the experiment.

Gagliano et al. performed two pilots: Stationary Light and Double Light Stationary Fan.

In stationary light, a LED was fixed on one arm and turned on for three 1-hr periods per day for 3 training days, then turned off for the testing day. All plants grew into the arm with the LED, confirming positive phototropism. In Double Light Stationary Fan, lights were affixed to both arms and a blowing fan was fixed on one arm. Plants grew at random, unaffected by the fan. Taken together, these pilots showed light is a suitable unconditioned stimulus and wind is a suitable neutral candidate conditioned stimulus, at least when both stimuli are stationary.

However, the main experiments involve moving stimuli. Therefore, we chose a different set of additional controls to replace the pilot studies, designated Moving Light, Moving Light Fan Opposite, and Moving Light Fan Adjacent. The training period for all three of these consisted of a light moving between maze arms in the same pattern as the lights in the four main conditions. On testing day, Moving Light was given no stimulus, Moving Light Fan Opposite had a fan placed opposite the last light exposure, and Moving Light Fan Adjacent had a fan placed on the same maze arm as the last light exposure. Moving Light is intended to confirm that the training regimen consistently results in growth towards the last presentation of light in the absence of wind, the other two are to check for systematic growth toward or away from the fan as well as any 'fan-induced amnesia,' wherein the fan interferes with the plant's growth towards the last presentation of light. If fans interfere with phototropic growth, this phenomenon would explain the Gagliano et al. results: phototropic growth without the fan on the testing day, random growth with the fan on testing day.

For the main experiment in both studies, seedlings were randomly assigned to two groups: F vs L presented light and fan on opposite maze arms, F + L presented light and fan on the same arm. The plants were trained for 3 days with 3 exposures per day with wind proceeding light by 60 min, overlapping for 30 min, then light only for an additional 30 min. These exposures were moved from one arm to the other according to the pattern: day 1, left (L)/right (R)/L; day 2, L/R/R; day 3 R/L/L, tested on R, or the inverse (plants were randomly assigned to opposite patterns and grouped for analysis). These 3 days of training were timed with the plants' growth such that on the fourth day the plants would grow into one arm or the other of the maze. Lights were disabled for this day, and the plants were subdivided into control and experimental groups. Control plants were left undisturbed and grew towards the arm where light was last presented on day 3, whereas experimental plants had fans placed so as to suggest light would be presented from the opposite arm. Gagliano et al. report that approximately two thirds of experimental plants showed behavior influenced by the fan, thus demonstrating associative learning. The setup and results of the two pilots and four main experimental conditions are shown in *Figure 1*.

The Gagliano et al. experiment was clever and innovative, and reports a finding of outstanding interest. Findings this unexpected have the potential to open up a new field of study, the establishment of which requires independent verification and experimental rigor (*Kuhn, 2012*). To that end, this study uses a larger sample size and fully blinded scoring.

Sample size in the Gagliano et al. study was sufficiently large to generate strong evidence, with several p values under 0.005. The present study increased sample size significantly to allow detection

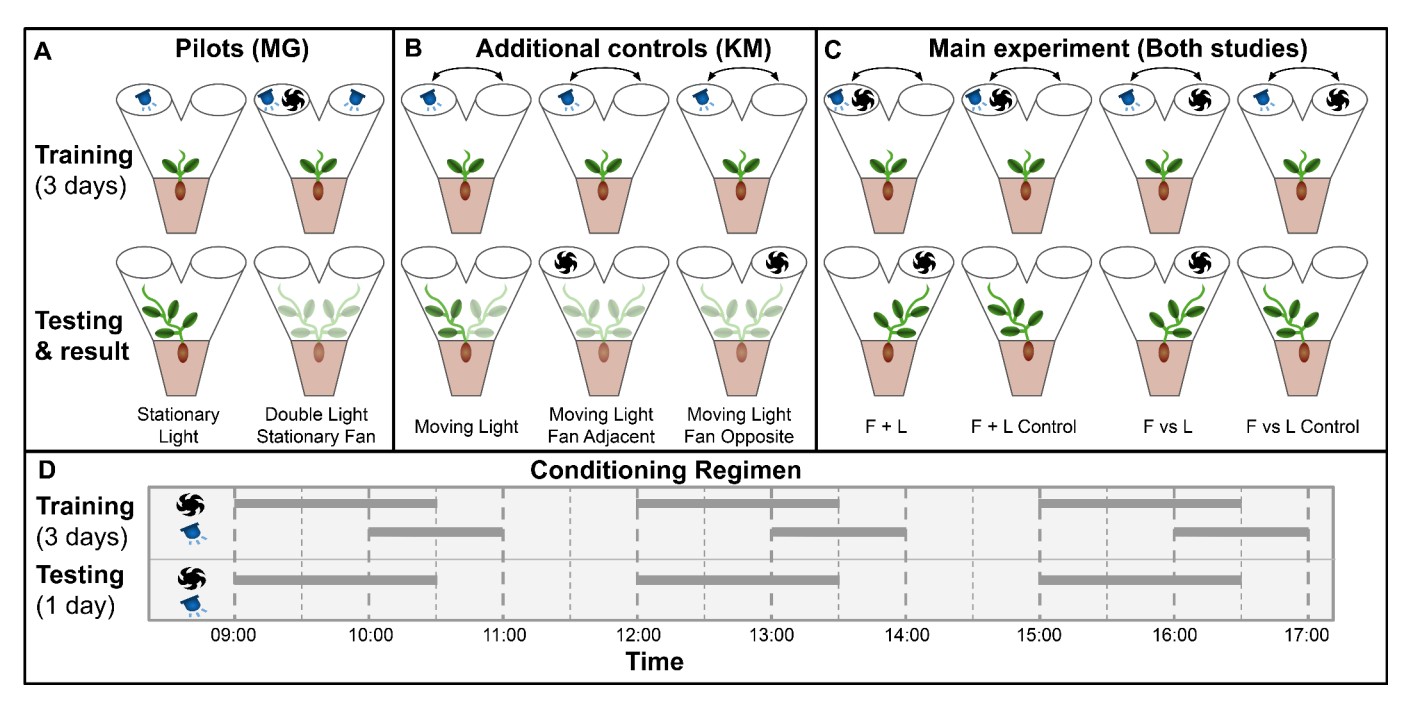

**Figure 1.** main Visual summary of all experimental conditions tested in both studies. (**A**) Pilot studies used in Gagliano et al. Top row shows maze fan and light configuration for training period, bottom row shows configuration during testing day and the maze arm into which plants grew. (**B**) Additional controls used in this study to address the same questions as the pilot studies in Gagliano et al. (**C**) Main experimental conditions.Results shown are from gagliano et al. in *Gagliano et al., 2016*. Results here are simplified, for more detailed results from both studies see *Figures 2* and *3*. (**D**) Conditioning regimen. Horizontal bars indicate time periods for which fans and lights were active, the x-axis indicates time of day.
The online version of this article includes the following figure supplement(s) for figure 1:

**Figure supplement 1.** Y maze detail.

of a weaker effect and estimate the effect size more precisely. The blinding protocol is also more complete. In the original study, the two experimenters also performed the scoring: the experimenter who set the experiment up recorded while the other scored the plants. In this replication attempt, scoring began with the experimenter removing all plants from the growth chamber, removing the fans and lights, and placing plants still within Y-mazes in a random order along the bench. Thereafter, an independent observer who had no other involvement in the experiment scored the plants and recorded scores while the experimenter was in a separate room. The scoring protocol was precisely defined as follows: if any part of the plants has grown more than 5 mm into a maze arm the plant was scored as choosing that arm, unless there was any growth into the other arm. If any growth was present in both arms or if <5 mm growth was present in either arm, plants were scored as choosing neither and not counted.

## Comparison of results

The additional controls Moving Light Fan Opposite and Moving Light Fan Adjacent were used to test for an inherent attraction or repulsion to the fan stimulus, and found no significant effect on growth direction (p=0.1713, Fisher's exact test [*Fisher, 1935*], two-tailed), as reported by Gagliano et al. This supports fans as an acceptable neutral stimulus. However, while Gagliano et al. demonstrated plants always grow toward a stationary light, we chose to ask whether plants always grow toward the last presentation of a light moving in the same regimen as the experiment, the Moving Light condition. Their growth did not differ significantly from a random 1:1 expectation at this sample size (p=0.5885, Fisher's exact test). The results of the additional control studies in this study and the Gagliano et al. pilots are compared in *Figure 2*.

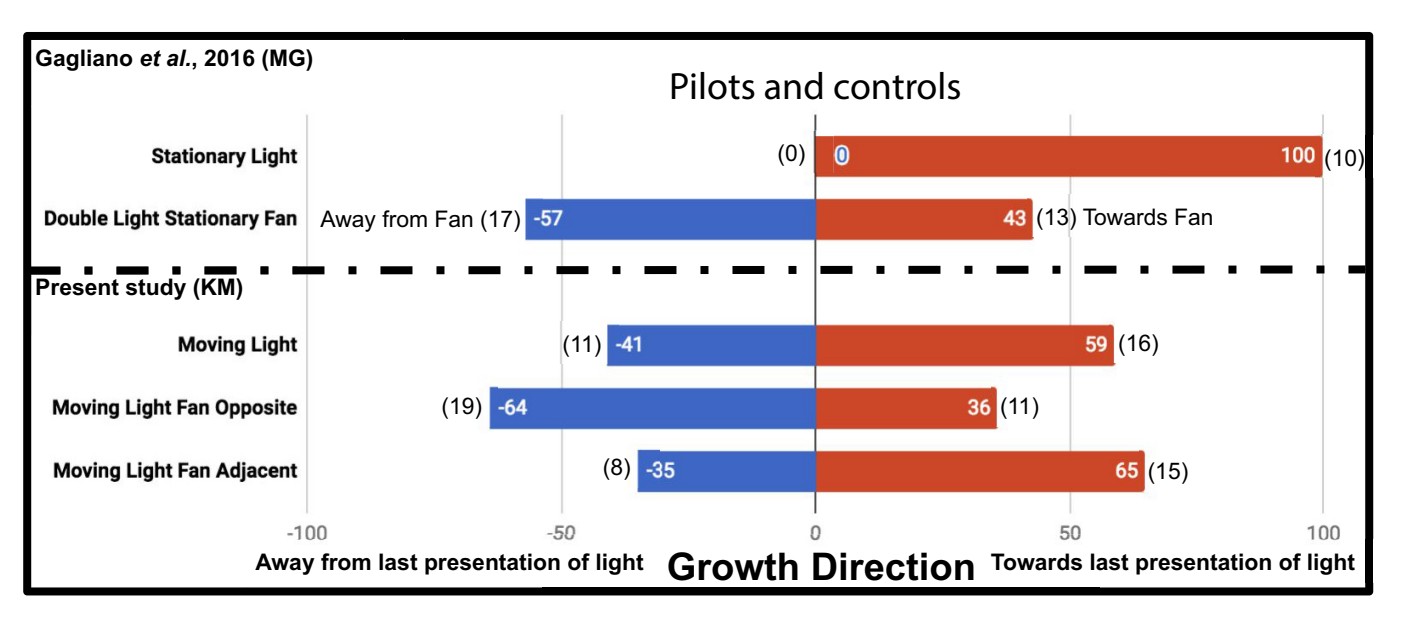

**Figure 2.** Results of plant growth in control conditions from the present study (KM) and the *Gagliano et al., 2016* experiment (MG). Numbers in bars indicate percentage, numbers in parentheses indicate raw data, number of plants growing into each maze arm.

In the four main conditions, the difference in results is fundamentally the same. F + L Control and F vs L Control plants did not always grow towards the last presentation of light, although interestingly both, like the Moving Light control, had a slight tendency in that direction. This tendency was statistically significant only in F + L control (also the condition with the largest sample size, n = 61, p=0.0243). In other words, there is a tendency to grow toward the last presentation of light, even if the presentation of light was very brief (1 hr) and proceeded by presentations of light from the opposite direction. However, we found this effect to be much weaker than the perfectly phototropic response Gagliano et al. report, despite using very similar lighting.

In the F + L Control and F vs L Control, we also found a slight tendency to grow toward the last presentation of light but far weaker than the perfect directionality reported by Gagliano et al. The noise in the control conditions makes any associative learning more difficult to detect. While Gagliano et al. report a significant difference between F + L and F + L Control (p=0.0027, n = 13 + 10, Fisher's exact test, two-tailed), we found no significant difference (p=0.335, n = 61 + 60). Similarly, Gagliano et al. report a significant difference between F vs L and F vs L Control (p=0.0017, n = 13 + 9), whereas we found no significant difference (p=0.387, n = 42 + 40). The results of the main four conditions are shown in *Figure 3*.

## Discussion

Gagliano et al. reported that in the absence of fan stimuli, pea plants always grow towards the last 1-hr presentation of light, whereas we find their growth to be only slightly biased toward the last presentation of light. It is possible that this difference is due to the use of different cultivars of *Pisum sativum* - Gagliano et al. used *cv* Massey gem, whereas this study used *cv* Green Arrow because Massey Gem was not available. These cultivars are closely related, are both full sun varieties, and have a similar growth habit. There were several other minor differences in materials used, but importantly growth conditions, maze construction, training regimen, and fan and light intensity were identical to the conditions reported by Gagliano et al. Because the capacity for associative learning is a complex trait, it is expected to not have evolved in the few decades since these cultivars shared a common ancestor, and seems equally unlikely to have been lost from one cultivar but not another in the absence of specific selection. However, a stronger phototropic response in Massey Gem cannot be excluded, calling for further replication efforts.

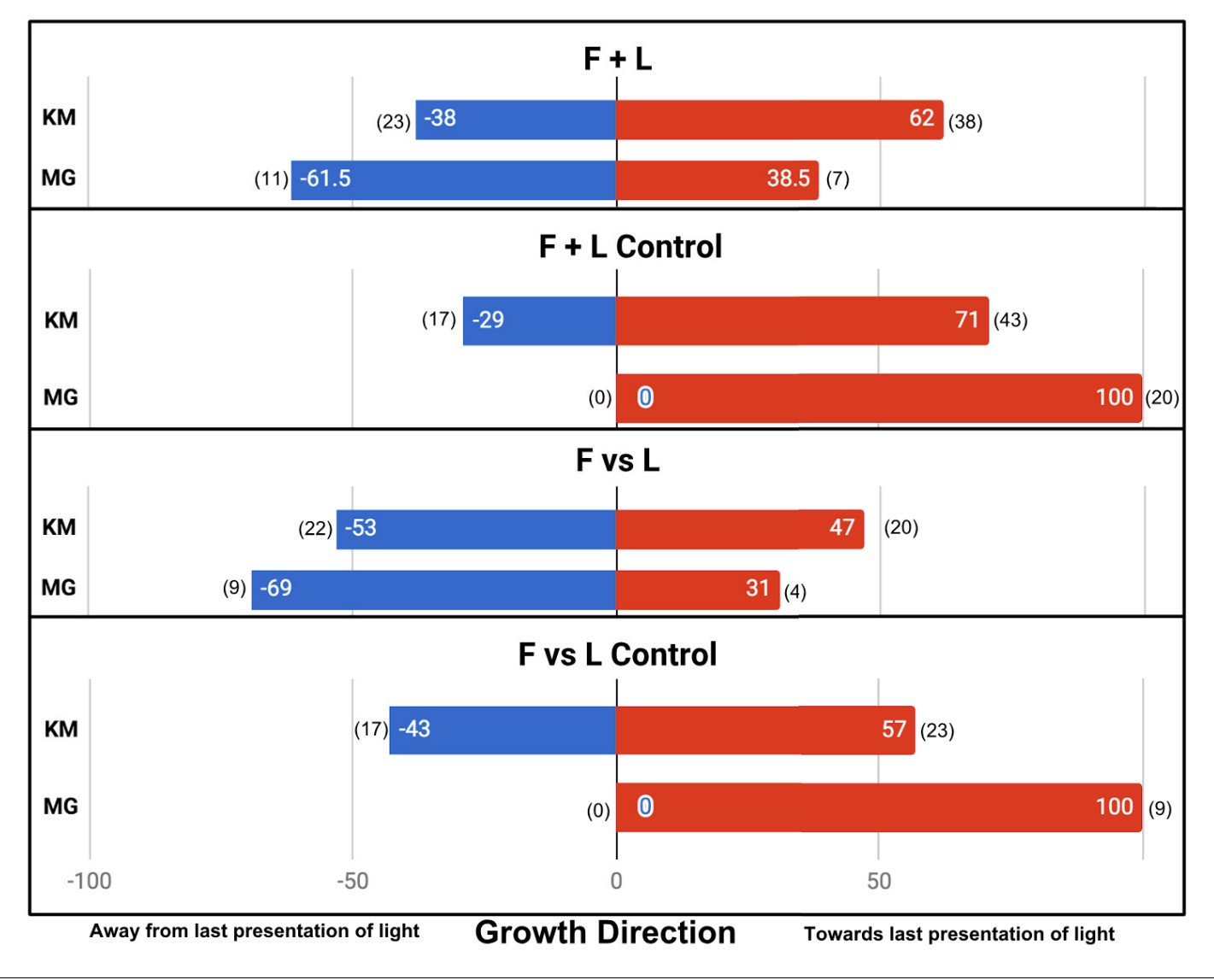

**Figure 3.** Results of plant growth in the four main conditions from this experiment (bars labeled KM) and the *Gagliano et al., 2016* paper (bars labeled MG). Numbers in parentheses indicate sample size. Data are pooled from the Gagliano et al. 'Experiment 1' and the Light group of their 'Experiment 2'.

We hope to see these performed using larger sample sizes and fully blinded scoring, and ideally with input and assistance from the original authors. Input from the original authors would allow for a more accurate replication, such as using the same pea plant supplier, LEDs, and fans, which may be critical to reproduce the phenomenon. If these efforts reproduce the phenomenon, we predict this experimental system will become widely used in experiments on plant learning. One shortcoming in the experimental system is that variation in plant growth rates results in substantial attrition. Plants were examined the morning of testing day to determine if they had already grown into one arm of the maze, and if they had they were excluded from the analysis. Furthermore, after testing day some plants had still not grown into either arm, and were therefore also excluded from the analysis. While care was taken to grow plants as uniformly as possible, approximately 40% of the plants were disqualified for growing too fast or too slow, as shown in *Table 1*.

The December 2016 report of associative learning in plants has garnered substantial attention in the press (*WNYC Studios, 2018*; *Morris, 2018*; *Berman, 2018*), and the reported phenomenon is extremely interesting. Gagliano et al. conclude this phenomenon will force us to reconsider the

**Table 1.** Sample size summary table.
This experiment (Columns labeled KM) suffered significant attrition due to variance in plant growth rate. Sample sizes from the *Gagliano et al., 2016* paper (MG) are shown for comparison, grouping results from their 'Experiment 1' and the light condition of 'Experiment 2'. Raw data available in *Source data 1*.

| Condition | Plants tested KM | Plants counted KM | Plants counted MG |
|---|---|---|---|
| Stationary Light | 0 | 0 | 20 |
| Double Light Stationary Fan | 0 | 0 | 30 |
| Moving Light | 50 | 27 | 0 |
| Moving Light Fan Opposite | 70 | 30 | 0 |
| Moving Light Fan Adjacent | 40 | 23 | 0 |
| F + L | 95 | 61 | 31 |
| F + L Control | 95 | 60 | 20 |
| F vs L | 65 | 42 | 13 |
| F vs L Control | 65 | 40 | 9 |

nature of learning (*Gagliano, 2017*) and we agree, contingent on the phenomenon being reproducible. Associative learning in the absence of traditional neurons must have a fascinating molecular mechanism and would open a new field of research. Unfortunately, no reports of successful replication have been published since the initial paper. Furthermore, this attempt at replication casts some doubt on the underlying premise used for scoring plant response, namely that plants will always grow towards the most recent one-hour light exposure after repeated exposures on different arms of a Y maze. At the least, it suggests that the conditions required for the experimental setup to function properly may be more precise than the stated parameters in the original paper. Additional work from the original authors and replications in independent labs are needed if this fascinating phenomenon is to move into the scientific mainstream.

## Materials and methods

### Germination conditions
Peas (*Pisum* sativum cv Green Arrow, Botanical Interests, USA) were germinated hydroponically in round containers kept in the dark. Seeds were first soaked in water for 24 hr, then wrapped by wet paper towel surrounded by aluminum foil in vertical rolls. These rolls were placed in water and incubated in the dark at 20°C, changing the water daily. Roughly three times as many seeds were germinated as plants desired, and after 5 days of water incubation seeds were inspected for germination using a dim red LED headlamp (~0.8 lux measured at light source,<0.2 lux measured at seedlings). Seeds with radicle >5 mm were considered to be germinated, ungerminated seeds were discarded. Among the germinated seeds, those with particularly long or short radicles were discarded to minimize variance in growth stage. Roughly twice times as many germinated seedlings were kept as plants desired. Each seedling was then planted in the center of round pots (5 cm diameter at top, 6 cm deep, 4 cm diameter at bottom, with a single drain hole of ~2 mm diameter in the center) in Hoffman seed starter potting and planting mix (Good Earth Inc, NY), at a depth of 15 mm. Soil was first watered to saturation, seeds were planted, and plants watered again to saturation. Soil was allowed to drain for ~30 min, then germinated seedlings were moved into a controlled growth chamber (PGR15 Growth Chamber, Conviron) until emergence from soil.

### Growth conditions
Chamber was maintained at 20°C, 85% humidity, with blue and red LEDs balanced for an approximation of white light. Light was delivered at 50 µmol m$^{-2}$ s$^{-1}$ at soil surface with 8:16 hr light:dark cycle with light phase beginning at 09:00 (identical to Gagliano et al.). After 3–4 days, most seedlings had emerged from soil. Seedlings at similar growth stages were selected, watered to saturation and

allowed to drain, and the Y maze attached. To correct for some plants being minorly off-center, all plants were rotated to maximize left-right symmetry before the attachment of the maze (importantly, the relevant growth directions had not been assigned at this stage, so there is no possible bias in the attachment of mazes, which were as centered as possible). Once all plants had mazes attached to their pots, they were assigned to groups using a random number generator, then grouped in rows and placed into the experimental growth chamber, where the fan and LEDs were attached on top of the maze arms.

## Apparatus details

The experimental apparatus consisted of blue LEDs with light intensity ~14 µmol m$^{-2}$ s$^{-1}$ (measured at soil surface) in the 430–505 nm wavelength range (CO RODE part#CR150514E156, Dr. Gagliano did not specify a particular LED model and did not respond to requests for model/part numbers). Fans were 35 mm 10,000 RPM computer cooling fans (Gdstime XH2.54-2Pin 3510S 35 × 10 mm brushless fan). The fans generated a semi-turbulent airflow of 0.6–0.8 m s$^{-1}$ at all points in the Y maze, including soil surface, the branching point of the maze, and on the fan arm and the opposite arm. Flow was approximately downward in the fan arm (parallel to the arm of the maze) and upward in the opposite arm, and was measured with TPI 575 anemometer hotwire probe and vane, which each gave similar results. These systems were soldered to two sets of 12V DC wires which extend throughout the growth chamber, and are powered by a digital automated timer system. 12V DC power was generated from 120 V 60 Hz AC power by a Winkeyes power supply (ASIN# B018G3ABWY).

## Scoring protocol

On testing day, during which plants must grow into one arm or the other, plants were further subdivided into control and experimental plants. All plants which had already grown into one maze arm were disqualified. Control plants were given no stimuli and left in the dark, experimental plants had the fan moved to the opposite arm from its last position and given the standard fan regimen without light. The day after testing day fans and lights were removed, plants were moved from growth chambers to a countertop and order-randomized before being scored by an independent observer with the experimenter in a separate room. Each plant was marked as either right, left, or neither. The criteria for was growth >5 mm above the decision point in only one maze arm. In the vast majority of cases, growth was in only one maze arm or neither, but two plants were disqualified and marked neither for growing into both maze arms.

## Data analysis

Once each plant's growth direction was recorded, plant numerical IDs were matched back to their experimental condition and enantiomeric pattern of stimulus exposure (with light beginning on either the left or the right maze arm), and scored one if their growth direction matched the direction of most recent light exposure and 0 if it was opposite the most recent light exposure. Scores were combined for each experimental condition and the proportion of plants growing according to the phototropic expectation was established and graphed. Fisher's exact test was used to determine different ratios between binary outcome between two conditions and is appropriate for small samples. This is the same analysis performed by Gagliano et al. All tests were performed with GraphPad QuickCalcs software.

## Sample size, attrition, and exclusion criteria

Gagliano et al. do not mention attrition or exclusion criteria in their report, but attrition in this study was quite high at several stages, the most problematic being due to variable growth rates resulting in plants reaching the decision point prior to the test day or failing to reach it 24 hr afterwards, both of which resulted in plants being disqualified. A smaller number of plants were disqualified because their individual light or fan system was found to have disconnected during movement from one maze arm to the other, therefore interrupting their training by missing part or all of a stimulus. In these cases, plants were disqualified by the experimenter during training phases and not scored. Number of plants trained and tested vs plants successfully scored as growing left or right are shown in *Table 1*. The controlled growth chamber used for training and testing plants had the capacity for

120 plants with mazes attached, and the experiment was performed four times in series to gather data. For each replication, approximately 350 seeds were germinated, of which approximately 250 would be planted, 120 transferred into the Y maze apparatus, and an average of 70.75 successfully scored as growing either left or right.

## Material differences between studies

Methods adapted from Gagliano et al., with minor modifications, mostly due to differences in product availability between the United States and Australia. Gagliano et al. used *Pisum* sativum cv Massey Gem (source unspecified), which was not available for my lab. Green Arrow (Botanical Interests, USA) was selected as a closely related cultivar with similar growth habit. I believe this difference should not affect the results, however, because associative learning must be a complex multigenic trait which is extremely unlikely to exist within one cultivar but not be present in a very closely related cultivar. Green Arrow, like Massey Gem, is a full sun cultivar, and thus should have similar propensity for phototropism. Similarly, Gagliano et al. used Osmocote seed raising and cutting mix, Scotts Australia), whereas I used Hoffman seed starter potting and planting mix (Good Earth Inc, NY), though its composition closely matches the soil used in the Gagliano et al. experiment. Instead of one 5.3 $m^2$ controlled environment room (source unspecified), two controlled growth chambers (PGR15 Growth Chamber, Conviron) were used, one for the initial growth and soil emergence stage, one for the training and testing stage. Importantly, the light, temperature, and humidity conditions in my chambers are the same as those of Gagliano et al. (see Germination Conditions and Growth Conditions), and the doors to the growth chambers were only opened with the outside room darkened to prevent the interference of external light.

## Acknowledgements

I thank Andrew Zink and Robyn Crook for their ideas regarding study design, Thomas Varley and Rebecca Dumanski for critical reading of the manuscript, Karen Markel for scoring plants, Anireddy Reddy for access to controlled growth chambers, Kyaw Tha Paw U for assistance with wind speed measurements, and Monica Gagliano for the original report and innovative experimental design.

## Additional information

### Funding

The author declares that there was no funding for this work.

### Author contributions

Kasey Markel, Resources, Data curation, Software, Formal analysis, Funding acquisition, Validation, Investigation, Visualization, Methodology, Writing - original draft, Project administration, Writing - review and editing

### Author ORCIDs

Kasey Markel https://orcid.org/0000-0002-8285-3888

### Decision letter and Author response

Decision letter https://doi.org/10.7554/eLife.57614.sa1
Author response https://doi.org/10.7554/eLife.57614.sa2

## Additional files

### Supplementary files

• Source data 1. Raw data for all experiments performed and statistical summaries of these data and *Gagliano et al., 2016* data.

• Transparent reporting form

## Data availability

All data generated or analyzed during this study are included in the manuscript and supporting files. Source data 1 contains all data collected, sufficient to reproduce all figures and statistical analyses.

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
