## [Decision Letter]

**Acceptance summary:**

This manuscript reports the results from a rigorous effort to replicate the findings from a recent study by Gagliano and colleagues (2016). While the results are negative, they raise major concerns about the extraordinary scientific claim by Gagliano et al. (2016), that associative learning is present in pea plants, which are devoid of a nervous system.

**Decision letter after peer review:**

Thank you for submitting your article "Pavlov's pea plants? Not so fast" for consideration by *eLife*. Your article has been reviewed by three peer reviewers, including Daeyeol Lee as the Reviewing Editor and Reviewer #1, and the evaluation has been overseen by Christian Hardtke as the Senior Editor The following individuals involved in review of your submission have agreed to reveal their identity: Jeansok J Kim (Reviewer #2); Stephanie Mary Groman (Reviewer #3).

The reviewers have discussed the reviews with one another and the Reviewing Editor has drafted this decision to help you prepare a revised submission.

Summary:

This manuscript reports the results from a failed effort to replicate the findings from a recent report by Gagliano et al. (2016) about associative learning in plants (pea). The study by Gagliano utilized a Pavlovian conditioning procedure using light and wind as unconditioned and conditioned stimuli, and showed that the plants tend to grow towards the conditioned stimuli after 3 days of conditioning. In the present study, despite the use of a larger sample size, the author failed to replicate this finding. Importantly, the author modified the control condition so that the light was no longer stationary, but changed its position each day, which made this control condition resemble the experimental conditions more. While the results are straightforward negative, they are nonetheless important because they address extraordinary scientific claims that learning is ubiquitous in all living organisms, including plants that have cellulose cell walls and devoid of nervous system. The manuscript is written very clearly, and there are no major concerns.

Essential revisions:

The rationale for using moving fans CS should be explained more clearly, since this is an important departure from Gagliano's original experiment. In addition, the organization of the paper in its current form makes it very difficult to understand how this experiment differs from that described in Gagliano. For example, the author first summarizes results of the original study and then describes what modifications were made to the current study. A more cohesive description might require the integration of these two sections – describe the procedures used in the current study and how the modifications control for aspects of the Gagliano study. In the same vein, Figure 1 is used to present the Gagliano design and results, but wouldn't it be more appropriate to include a figure of the design of the current manuscript? This would be a good place to provide a visual description of each of the conditions using the same acronyms/terms used in the manuscript.

---

## [Author Response]

Essential revisions:The rationale for using moving fans CS should be explained more clearly, since this is an important departure from Gagliano's original experiment.

The conditioned stimulus is the same in both studies, moving fan. In both studies, only during the four ‘main conditions’ is associative conditioning expected to take place. The departure is between Gagliano’s pilots and my ‘additional controls.’ I am using a moving light in these controls because a moving light is used in the four main experimental conditions.

However, in retrospect I should also have included a stationary light condition – in fact I did do so as a pilot, but due to technical difficulties with the then-new experimental setup the data were untrustworthy, and I didn’t repeat the experiment. This would allow for a better understanding of why my plants had a much lower rate of phototropic growth than Gagliano et al.

In addition, the organization of the paper in its current form makes it very difficult to understand how this experiment differs from that described in Gagliano. For example, the author first summarizes results of the original study and then describes what modifications were made to the current study. A more cohesive description might require the integration of these two sections – describe the procedures used in the current study and how the modifications control for aspects of the Gagliano study.

These sections have been integrated. I hope the differences between the two studies are more clear.

In the same vein, Figure 1 is used to present the Gagliano design and results, but wouldn't it be more appropriate to include a figure of the design of the current manuscript? This would be a good place to provide a visual description of each of the conditions using the same acronyms/terms used in the manuscript.

Excellent point. Figure 1 has been expanded to show all conditions tested in both studies. The most important four conditions are shared in both studies, as shown in Figure 1C. Gagliano et al.’s two pilots and my three ‘additional controls’ are shown in Figure 1A and B, respectively. This should provide a good visual explanation of the names of conditions.